# A MAX-AFFINE SPLINE PERSPECTIVE OF RECURRENT NEURAL NETWORKS

**Zichao Wang, Randall Balestriero & Richard G. Baraniuk**
Department of Electrical and Computer Engineering
Rice University
Houston, TX 77005, USA
`{zw16,rb42,richb}@rice.edu`

## ABSTRACT

We develop a framework for understanding and improving recurrent neural networks (RNNs) using *max-affine spline operators* (MASOs). We prove that RNNs using piecewise affine and convex nonlinearities can be written as a simple piecewise affine spline operator. The resulting representation provides several new perspectives for analyzing RNNs, three of which we study in this paper. First, we show that an RNN internally partitions the input space during training and that it builds up the partition through time. Second, we show that the affine slope parameter of an RNN corresponds to an input-specific template, from which we can interpret an RNN as performing a simple template matching (matched filtering) given the input. Third, by carefully examining the MASO RNN affine mapping, we prove that using a *random initial hidden state* corresponds to an explicit $\ell_2$ regularization of the affine parameters, which can mollify exploding gradients and improve generalization. Extensive experiments on several datasets of various modalities demonstrate and validate each of the above conclusions. In particular, using a random initial hidden states elevates simple RNNs to near state-of-the-art performers on these datasets.

## 1 INTRODUCTION

Recurrent neural networks (RNNs) are a powerful class of models for processing sequential inputs and a basic building block for more advanced models that have found success in challenging problems involving sequential data, including sequence classification (e.g., sentiment analysis (Socher et al., 2013) , sequence generation (e.g., machine translation (Bahdanau et al., 2014)), speech recognition (Graves et al., 2013), and image captioning (Mao et al., 2015). Despite their success, however, our understanding of how RNNs work remains limited. For instance, an attractive theoretical result is the universal approximation property that states that an RNN can approximate an arbitrary function (Schäfer & Zimmermann, 2006; Siegelmann & Sontag, 1995; Hammer, 2000). These classical theoretical results have been obtained primarily from the dynamical system (Siegelmann & Sontag, 1995; Schäfer & Zimmermann, 2006) and measure theory (Hammer, 2000) perspectives. These theories provide approximation error bounds but unfortunately limited guidance on applying RNNs and understanding their performance and behavior in practice.

In this paper, we provide a new angle for understanding RNNs using *max-affine spline operators* (MASOs) (Magnani & Boyd, 2009; Hannah & Dunson, 2013) from approximation theory. The piecewise affine approximations made by compositions of MASOs provide a new and useful framework to study neural networks. For example, Balestriero & Baraniuk (2018); Balestriero & Baraniuk (2018a) have provided a detailed analysis in the context of feedforward networks. Here, we go one step further and find new insights and interpretations from the MASO perspective for RNNs. We will see that the input space partitioning and matched filtering links developed in Balestriero & Baraniuk (2018); Balestriero & Baraniuk (2018a) extend to RNNs and yield interesting insights into their inner workings. Moreover, the MASO formulation of RNNs enables us to theoretically justify the use of a random initial hidden state to improve RNN performance.

For concreteness, we focus our analysis on a specific class of simple RNNs (Elman, 1990) with *piecewise affine and convex nonlinearities* such as the ReLU (Glorot et al., 2011). RNNs with

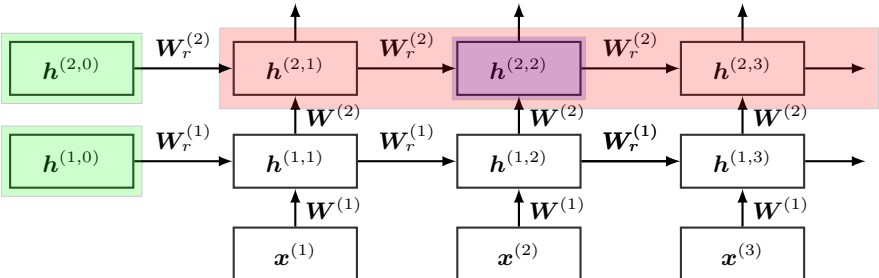

Figure 1: Visualization of an RNN that highlights a cell (purple), a layer (red) and the initial hidden state of each layer (green). (Best viewed in color.)

such nonlinearities have recently gained considerable attention due to their ability to combat the *exploding gradient problem*; with proper initialization (Le et al., 2015; Talathi & Vartak, 2016) and clever parametrization of the recurrent weight (Arjovsky et al., 2016; Wisdom et al., 2016; Jing et al., 2017; Hyland & Rätsch, 2017; Mhammedi et al., 2017; Helfrich et al., 2018), these RNNs achieve performance on par with more complex ones such as LSTMs. Below is a summary of our key contributions.

**Contribution 1.** We prove that an RNN with piecewise affine and convex nonlinearities can be rewritten as a composition of MASOs, making it a piecewise affine spline operator with an elegant analytical form (Section 3).

**Contribution 2.** We leverage the partitioning of piecewise affine spline operators to analyze the input space partitioning that an RNN implicitly performs. We show that an RNN calculates a new, high-dimensional representation (the partition code) of the input sequence that captures informative underlying characteristics of the input. We also provide a new perspective on RNN dynamics by visualizing the evolution of the RNN input space partitioning through time (Section 4).

**Contribution 3.** We show the piecewise affine mapping in an RNN associated with a given input sequence corresponds to an input-dependent template, from which we can interpret the RNN as performing greedy template matching (matched filtering) at every RNN cell (Section 5).

**Contribution 4.** We rigorously prove that using a random (rather than zero) initial hidden state in an RNN corresponds to an explicit regularizer that can mollify exploding gradients. We show empirically that such a regularization improves RNN performance (to state-of-the-art) on four datasets of different modalities (Section 6).

## 2 BACKGROUND

**Recurrent Neural Networks (RNNs).** A simple RNN unit (Elman, 1990) per layer $\ell$ and time step $t$, referred to as a "cell," performs the following recursive computation

$$\boldsymbol{h}^{(\ell,t)} = \sigma\left(\boldsymbol{W}^{(\ell)}\boldsymbol{h}^{(\ell-1,t)} + \boldsymbol{W}_r^{(\ell)}\boldsymbol{h}^{(\ell,t-1)} + \boldsymbol{b}^{(\ell)}\right), \tag{1}$$

where $\boldsymbol{h}^{(\ell,t)}$ is the hidden state at layer $\ell$ and time step $t$, $\boldsymbol{h}^{(0,t)} := \boldsymbol{x}^{(t)}$ which is the input sequence, $\sigma$ is an activation function and $\boldsymbol{W}^{(\ell)}, \boldsymbol{W}_r^{(\ell)}$, and $\boldsymbol{b}^{(\ell)}$ are time-invariant parameters at layer $\ell$. $\boldsymbol{h}^{(\ell,0)}$ is the initial hidden state at layer $\ell$ which needs to be set to some value beforehand to start the RNN recursive computation. Unrolling the RNN through time gives an intuitive view of the RNN dynamics, which we visualize in Figure 1. The output of the overall RNN is typically an affine transformation of the hidden state of the last layer $L$ at time step $t$

$$\boldsymbol{z}^{(t)} = \boldsymbol{W}\boldsymbol{h}^{(L,t)} + \boldsymbol{b}. \tag{2}$$

In the special case where the RNN has only one output at the end of processing the entire input sequence, the RNN output is an affine transformation of the hidden state at the last time step, i.e., $\boldsymbol{z}^{(T)} = \boldsymbol{W}\boldsymbol{h}^{(L,T)} + \boldsymbol{b}$.

**Max-Affine Spline Operators (MASOs).** A max-affine spline operator (MASO) is piecewise affine and convex with respect to each output dimension $k = 1, \ldots, K$. It is defined as a parametric function $S : \mathbb{R}^D \to \mathbb{R}^K$ with parameters $\boldsymbol{A} \in \mathbb{R}^{K \times R \times D}$ and $\boldsymbol{B} \in \mathbb{R}^{K \times R}$. A MASO leverages $K$

independent max-affine splines (Magnani & Boyd, 2009), each with $R$ partition regions. Its output for output dimension $k$ is produced via

$$[\boldsymbol{y}]_k = [S(\boldsymbol{x})]_k = \max_{r=1,\dots,R} \left\{ \langle [\boldsymbol{A}]_{k,r,\cdot}, \boldsymbol{x} \rangle + [\boldsymbol{B}]_{k,r} \right\}, \tag{3}$$

where $\boldsymbol{x} \in \mathbb{R}^D$ and $\boldsymbol{y} \in \mathbb{R}^K$ are dummy variables that respectively denote the input and output of the MASO $S$ and $\langle \cdot, \cdot \rangle$ denotes inner product. The three subscripts of the "slope" parameter $[\boldsymbol{A}]_{k,r,d}$ correspond to output $k$, partition region $r$, and input signal index $d$. The two subscripts of the "bias" parameter $[\boldsymbol{B}]_{k,r}$ correspond to output $k$ and partition region $r$.

We highlight two important and interrelated MASO properties relevant to the discussions throughout the paper. First, a MASO performs implicit input space partitioning, which is made explicit by rewriting (3) as

$$[\boldsymbol{y}]_k = \sum_{r=1}^{R} [Q]_{k,r} \left( \langle [\boldsymbol{A}]_{k,r,\cdot}, \boldsymbol{x} \rangle + [\boldsymbol{B}]_{k,r} \right), \tag{4}$$

where $Q \in \mathbb{R}^{K \times R}$ is a partition selection matrix[1] calculated as

$$[Q]_{k,r} = \mathbb{1}(r = [\boldsymbol{r}^*]_k), \quad \text{where} \ \ [\boldsymbol{r}^*]_k = \arg\max_{r=1,\cdots,R} \langle [\boldsymbol{A}]_{k,r,\cdot}, \boldsymbol{x} \rangle + [\boldsymbol{B}]_{k,r}. \tag{5}$$

Namely, $Q$ contains $K$ stacked one-hot row vectors, each of which selects the $[\boldsymbol{r}^*]_k^{\text{th}}$ partition of the input space that maximizes (4) for output dimension $k$. As a consequence, knowing $Q$ is equivalent to knowing the partition of an input $\boldsymbol{x}$ that the MASO implicitly computes. We will use this property in Section 4 to provide new insights into RNN dynamics.

Second, given the partition $\boldsymbol{r}^*$ that an input belongs to, as determined by (5), the output of the MASO of dimension $k$ from (3) reduces to a simple affine transformation of the input

$$[\boldsymbol{y}]_k = [A]_{k,\cdot} \boldsymbol{x} + [B]_k, \quad \text{where} \ \ [A]_{k,\cdot} = [\boldsymbol{A}]_{k,[\boldsymbol{r}^*]_k} \ \text{and} \ [B]_{k,\cdot} = [\boldsymbol{B}]_{k,[\boldsymbol{r}^*]_k}. \tag{6}$$

Here, the selected affine parameters $A \in \mathbb{R}^{K \times D}$ and $B \in \mathbb{R}^K$ are specific to the input's partition region $[\boldsymbol{r}^*]_k$, which are simply the $[\boldsymbol{r}^*]_k^{\text{th}}$ slice and $[\boldsymbol{r}^*]_k^{\text{th}}$ column of $\boldsymbol{A}$ and $\boldsymbol{B}$, respectively, for output dimension $k$. We emphasize that $A$ and $B$ are input-dependent; different inputs $\boldsymbol{x}$ induce different $A$ and $B$.[2] We will use this property in Section 5 to link RNNs to matched filterbanks.

## 3 RNNs as Piecewise Affine Spline Operators

We now leverage the MASO framework to rewrite, interpret, and analyze RNNs. We focus on RNNs with piecewise affine and convex nonlinearities in order to derive rigorous analytical results. The analysis of RNNs with other nonlinearities is left for future work.

We first derive the MASO formula for an RNN cell (1) and then extend to one layer of a time-unrolled RNN and finally to a multi-layer, time-unrolled RNN. Let $\boldsymbol{z}^{(\ell,t)} = \left[ \boldsymbol{h}^{(\ell-1,t)^\top}, \boldsymbol{h}^{(\ell,t-1)^\top} \right]^\top$ be the input to an RNN cell that is the concatenation of the current input $\boldsymbol{h}^{(\ell-1,t)}$ and the previous hidden state $\boldsymbol{h}^{(\ell,t-1)}$. Then we have the following result, which is a straightforward extension of Proposition 4 in Balestriero & Baraniuk (2018).

**Proposition 1.** *An RNN cell of the form (1) is a MASO with*

$$\boldsymbol{h}^{(\ell,t)} = A^{(\ell,t)} \boldsymbol{z}^{(\ell,t)} + B^{(\ell,t)}, \tag{7}$$

*where $A^{(\ell,t)} = A_\sigma^{(\ell,t)} [\boldsymbol{W}^{(\ell)}, \boldsymbol{W}_r^{(\ell)}]$ and $B^{(t)} = A_\sigma^{(\ell,t)} \boldsymbol{b}^{(\ell)}$ are the affine parameters and $A_\sigma^{(\ell,t)}$ is the affine parameter corresponding to the piecewise affine and convex nonlinearity $\sigma(\cdot)$ that depends on the cell input $\boldsymbol{z}^{(\ell,t)}$.*

---

[1]Prior work denotes the partition selection matrix as $T$. But in the context of RNNs, $T$ usually denotes the length of the input sequence. Thus we denote this matrix as $Q$ in this work to avoid notation conflicts.

[2]The notation for the affine spline parameters $A$ and $B$ in (Balestriero & Baraniuk, 2018; Balestriero & Baraniuk, 2018b) are $A[\boldsymbol{x}]$ and $B[\boldsymbol{x}]$, respectively, in order to highlight their input dependency. In this paper, we drop the input dependency when writing these affine parameters to simplify the notation, and we use brackets to exclusively denote indexing or concatenation.

We now derive an explicit affine formula for a time-unrolled RNN at layer $\ell$. Let $\boldsymbol{h}^{(\ell-1)} = \left[\boldsymbol{h}^{(\ell-1,1)^\top}, \cdots, \boldsymbol{h}^{(\ell-1,T)^\top}\right]^\top$ be the entire input sequence to the RNN at layer $\ell$, and let $\boldsymbol{h}^{(\ell)} = \left[\boldsymbol{h}^{(\ell,1)^\top}, \cdots, \boldsymbol{h}^{(\ell,T)^\top}\right]^\top$ be all the hidden states that are output at layer $\ell$. After some algebra and simplification, we arrive at the following result.

**Theorem 1.** *The $\ell^{\text{th}}$ layer of an RNN is a piecewise affine spline operator defined as*

$$
\begin{pmatrix} \boldsymbol{h}^{(\ell,T)} \\ \vdots \\ \boldsymbol{h}^{(\ell,1)} \end{pmatrix} = \underbrace{\begin{pmatrix} \mathcal{A}_{T:T}^{(\ell)} \cdots \mathcal{A}_{1:T}^{(\ell)} \\ \vdots \ddots \vdots \\ \boldsymbol{0} \cdots \mathcal{A}_{1:1}^{(\ell)} \end{pmatrix}}_{\text{upper triangular}} \underbrace{\begin{pmatrix} A_\sigma^{(\ell,T)} \boldsymbol{W}^{(\ell)} \cdots \boldsymbol{0} \\ \vdots \ddots \vdots \\ \boldsymbol{0} \cdots A_\sigma^{(\ell,1)} \boldsymbol{W}^{(\ell)} \end{pmatrix}}_{\text{diagonal}} \begin{pmatrix} \boldsymbol{h}^{(\ell-1,T)} \\ \vdots \\ \boldsymbol{h}^{(\ell-1,1)} \end{pmatrix}
$$

$$
+ \begin{pmatrix} \sum\limits_{t=T}^{1} \mathcal{A}_{t:T}^{(\ell)} B^{(\ell,t)} + \mathcal{A}_{0:T}^{(\ell)} \boldsymbol{h}^{(\ell,0)} \\ \vdots \\ \mathcal{A}_{1:1}^{(\ell)} B^{(\ell,t)} + \mathcal{A}_{0:1}^{(\ell)} \boldsymbol{h}^{(\ell,0)} \end{pmatrix} = A_{\text{RNN}}^{(\ell)} \boldsymbol{h}^{(\ell-1)} + B_{\text{RNN}}^{(\ell)}, \quad (8)
$$

*where $\mathcal{A}_{t:T'}^{(\ell)} = \left(\prod_{s=T'}^{t+1} A_\sigma^{(\ell,s)} \boldsymbol{W}_r^{(\ell)}\right)$ for $t < T'$ and identity otherwise,[3] $\boldsymbol{h}^{(\ell,0)}$ is the initial hidden state of the RNN at layer $\ell$, and $A_{\text{RNN}}^{(\ell)}$ and $B_{\text{RNN}}^{(\ell)}$ are affine parameters that depend on the layer input $\boldsymbol{h}^{(\ell-1)}$ and the initial hidden state $\boldsymbol{h}^{(\ell,0)}$.*

We present the proof for Theorem 1 in Appendix G. The key point here is that, by leveraging MASOs, we can represent the time-unrolled RNN as a simple affine transformation of the entire input sequence (8). Note that this affine transformation changes depending on the partition region in which the input belongs (recall (4) and (5)). Note also that the initial hidden state affects the layer output by influencing the affine parameters and contributing a bias term $A_{0:t}^{(\ell)} \boldsymbol{h}^{(\ell,0)}$ to the bias parameter $B_{\text{RNN}}^{(\ell)}$. We study the impact of the initial hidden state in more detail in Section 6.

We are now ready to generalize the above result to multi-layer RNNs. Let $\boldsymbol{x} = \left[\boldsymbol{x}^{(1)^\top}, \cdots, \boldsymbol{x}^{(T)^\top}\right]^\top$ be the input sequence to a multi-layer RNN, and let $\boldsymbol{z} = \left[\boldsymbol{z}^{(1)^\top}, \cdots, \boldsymbol{z}^{(T)^\top}\right]^\top$ be the output sequence. We state the following result for the overall mapping of a multi-layer RNN.

**Theorem 2.** *The output of an L-layer RNN is a piecewise affine spline operator defined as*

$$
\boldsymbol{z} = \mathcal{W}\left(\boldsymbol{h}^{(L)}\right) + \boldsymbol{b} = \mathcal{W}\left(A_{\text{RNN}} \boldsymbol{x} + B_{\text{RNN}}\right) + \boldsymbol{b}, \quad (9)
$$

*where $A_{\text{RNN}} = \prod_{\ell=L}^{1} A_{\text{RNN}}^{(\ell)}$ and $B_{\text{RNN}} = \sum_{\ell=1}^{L} \left(\prod_{\ell'=\ell}^{L-1} A_{\text{RNN}}^{(\ell')}\right) B_{\text{RNN}}^{(\ell)}$ are the affine parameters of the L-layer RNN. $\mathcal{W}$ and $\boldsymbol{b}$ are parameters of the fully connected output layer, where $\mathcal{W} = [\boldsymbol{W}, \boldsymbol{W}, \ldots, \boldsymbol{W}]$ when the RNN outputs at every time step and $\mathcal{W} = [\boldsymbol{W}, 0, \ldots, 0]$ when the RNN outputs only at the last time step.*

Theorem 2 shows that, using MASOs, we have a simple, elegant, and closed-form formula showing that the output of an RNN is computed locally via very simple functions. This result is proved by recursively applying the proof for Theorem 1.

The affine mapping formula (9) opens many doors for RNN analyses, because we can shed light on RNNs by applying established matrix results. In the next sections, we provide three analyses that follow this programme. First, we show that RNNs partition the input space and that they develop the partitions through time. Second, we analyze the forms of the affine slope parameter and link RNNs to matched filterbanks. Third, we study the impact of the initial hidden state to rigorously justify the use of randomness in initial hidden state. From this point, for simplicity, we will assume a zero initial hidden state unless otherwise stated.

## 4 Internal Input Space Partitioning in RNNs

The MASO viewpoint enables us to see how an RNN implicitly partitions its input sequence through time, which provides a new perspective of its dynamics. To see this, first recall that, for an RNN

---

[3]In our context, $\prod_{i=m}^{n} a_i := a_m \cdot a_{m-1} \cdots a_{n+1} \cdot a_n$ for $m > n$ as opposed to the empty product.

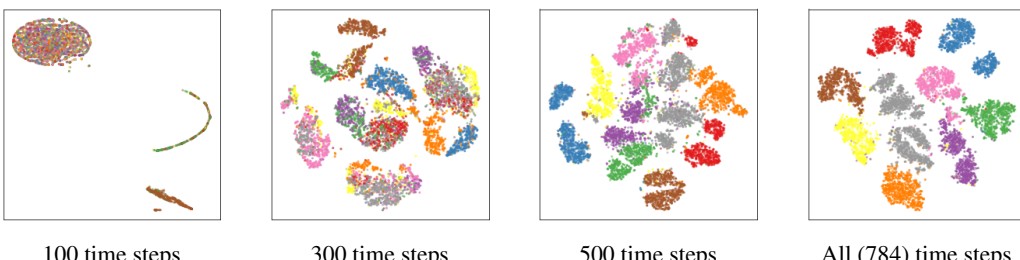

100 time steps          300 time steps          500 time steps          All (784) time steps

Figure 2: t-SNE (van der Maaten & Hinton, 2008) visualization of the evolution of the RNN partition codes of input sequences from the MNIST test set. Each color represents one of the ten classes. We see clearly that the RNN gradually develops and refines the partition codes through time to separate the classes.

cell, the piecewise affine and convex activation nonlinearity partitions each dimension of the cell input $z^{(\ell,t)}$ into $R$ regions (for ReLU, $R = 2$). Knowing the state of the nonlinearity (which region $r$ is activated) is thus equivalent to knowing the partition of the cell input. For a multi-layer RNN composed of many RNN cells (recall Figure 1), the input sequence partition can be retrieved by accessing the collection of the states of all of the nonlinearities; each input sequence can be represented by a partition "code" that determines the partition to which it belongs.

Since an RNN processes an input sequence one step at a time, the input space partition is gradually built up and refined through time. As a consequence, when seen through the MASO lens, the forward pass of an RNN is simply developing and refining the partition code of the input sequence. Visualizing the evolution of the partition codes can be potentially beneficial for diagnosing RNNs and understanding their dynamics.

As an example, we demonstrate the evolution of the partition codes of a one-layer ReLU RNN trained on the MNIST dataset, with each image flattened into a 1-dimensional sequence so that input at each time step is a single pixel. Details of the model and experiments are in Appendix C. Since the ReLU activation partitions its input space into only 2 regions , we can retrieve the RNN partition codes of the input images simply by binarizing and concatenating all of the hidden states. Figure 2 visualizes how the partition codes of MNIST images evolve through time using t-SNE, a distance-preserving dimensionality reduction technique (van der Maaten & Hinton, 2008). The figure clearly shows the evolution of the partition codes from hardly any separation between classes of digits to forming more and better separated clusters through time. We can also be assured that the model is well-behaved, since the final partition shows that the images are well clustered based on their labels. Additional visualizations are available in Section D.

## 5   RNNs as Matched Filterbanks

The MASO viewpoint enables us to connect RNNs to classical signal processing tools like the matched filter. Indeed, we can directly interpret an RNN as a *matched filterbank*, where the classification decision is informed via the simple inner product between a "template" and the input sequence. To see this, we follow an argument similar to that in Section 4. First, note that the slope parameter $A^{(\ell,t)}$ for each RNN cell is a "locally optimal template" because it maximizes each of its output dimensions over the $R$ regions that the nonlinearity induces (recall (3) and (7)). For a multi-layer RNN composed of many RNN cells, the overall "template" $A_{\mathrm{RNN}}$ corresponds to the composition of the optimal templates from each RNN cell, which can be computed simply via $\mathrm{d}z/\mathrm{d}x$ (recall (9)).

Thus, we can view an RNN as a matched filterbank whose output is the maximum inner product between the input and the rows of the overall template $A_{\mathrm{RNN}}$ (van Trees, 1992; 2013). The overall template is also known in the machine learning community as a *salience map*; see Li et al. (2016) for an example of using saliency maps to visualize RNNs. Our new insight here is that a good template produces a larger inner product with the input *regardless of the visual quality of the template*, thus complementing prior work. The template matching view of RNNs thus provides a principled methodology to visualize and diagnose RNNs by examining the inner products between the inputs and the templates.

To illustrate the matched filter interpretation, we train a one-layer ReLU RNN on the polarized Stanford Sentiment Treebank dataset (SST-2) (Socher et al., 2013), which poses a binary classification

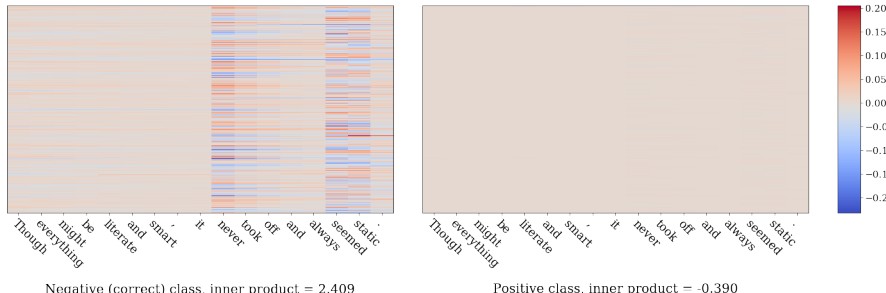

Figure 3: Templates corresponding to the correct (left) and incorrect class (right) of a negative sentiment input from the SST-2 dataset. Each column contains the gradient corresponding to an input word. Quantitatively, we can see that the inner product between input and the correct class template (left) produces a larger value than that between input and the incorrect class template (right).

problem, and display in Figure 3 the templates corresponding to the correct and incorrect classes of an input where the correct class is a negative sentiment. We see that the input has a much larger inner product with the template corresponding to the correct class (left plot) than that corresponding to the incorrect class (right plot), which informs us that the model correctly classifies this input. Additional experimental results are given in Appendix E.

## 6 IMPROVING RNNs VIA RANDOM INITIAL HIDDEN STATE

In this section, we provide a theoretical motivation for the use of a random initial hidden state in RNNs. The initial hidden state needs to be set to some prior value to start the recursion (recall Section 2). Little is understood regarding the best choice of initial hidden state other than Zimmermann et al. (2012)'s dynamical system argument. Consequently, it is typically simply set to zero. Leveraging the MASO view of RNNs, we now demonstrate that one can improve significantly over a zero initial hidden state by using a *random initial hidden state*. This choice regularizes the affine slope parameter associated with the initial hidden state and mollifies the so-called exploding gradient problem (Pascanu et al., 2013).

**Random Initial Hidden State as an Explicit Regularization.** We first state our theoretical result that using random initial hidden state corresponds to an explicit regularization and then discuss its impact on exploding gradients. Without loss of generality, we focus on one-layer ReLU RNNs. Let $N$ be the number of data points and $C$ the number of classes. Define $\mathcal{A}_h := \mathcal{A}_{1:T} = \prod_{s=T}^{1} A_\sigma^{(s)} \boldsymbol{W}_r^{(\ell)}$ (recall (8)).

**Theorem 3.** *Let $\mathcal{L}$ be an RNN loss function, and let $\widetilde{\mathcal{L}}$ represent the modified loss function when the RNN initial hidden state is set to a Gaussian random vector $\epsilon \sim \mathcal{N}(0, \sigma_\epsilon^2 \boldsymbol{I})$ with small standard deviation $\sigma_\epsilon$. Then we have that $\mathbb{E}_\epsilon\left[\widetilde{\mathcal{L}}\right] = \mathcal{L} + \mathcal{R}$. For the cross-entropy loss $\mathcal{L}$ with softmax output,*

$$\mathcal{R} = \frac{\sigma_\epsilon^2}{2N} \sum_{n=1}^{N} \left\| \operatorname{diag}\left(\left[\frac{\mathrm{d}\widehat{y}_{ni}}{\partial z_{nj}}\right]_{i=j}\right) \mathcal{A}_h \right\|^2,$$ *where $\widehat{y}_{ni}$ is the $i^{\text{th}}$ dimension of the softmax output of the $n^{\text{th}}$ data point and $i, j \in \{1, \ldots, C\}$ are the class indices. For the mean-squared error loss $\mathcal{L}$, $\mathcal{R} = \frac{\sigma_\epsilon^2}{2N} \sum_{n=1}^{N} \|\mathcal{A}_h\|^2$.*

We prove this result for the cross-entropy loss in Appendix G.2. The standard deviation $\sigma_\epsilon$ controls the importance of the regularization term and recovers the case of standard zero initial hidden state when $\sigma_\epsilon = 0$.

**Connection to the Exploding Gradient Problem.** Backpropagation through time (BPTT) is the default RNN training algorithm. Updating the recurrent weight $\boldsymbol{W}_r$ with its gradient using BPTT involves calculating the gradient of the RNN output with respect to the hidden state at each time step $t = 0, \ldots, T$

$$\frac{\mathrm{d}\mathcal{L}}{\mathrm{d}\boldsymbol{h}^{(t)}} = \frac{\mathrm{d}\mathcal{L}}{\mathrm{d}\boldsymbol{z}} \frac{\mathrm{d}\boldsymbol{z}}{\mathrm{d}\boldsymbol{h}^{(T)}} \left(\prod_{s=T}^{t+1} \frac{\mathrm{d}\boldsymbol{h}^{(T)}}{\mathrm{d}\boldsymbol{h}^{(s)}}\right) = \frac{\mathrm{d}\mathcal{L}}{\mathrm{d}\boldsymbol{z}} \boldsymbol{W} \left(\prod_{s=T}^{t+1} A_\sigma^{(s)} \boldsymbol{W}_r\right). \tag{10}$$

When $\|A_\sigma^{(s)} \boldsymbol{W}_r\|_2 > 1$, the product term $\prod_{s=T}^{t+1} A_\sigma^{(s)} \boldsymbol{W}_r$ in (10) blows up, which leads to unstable training. This is known as the *exploding gradient problem* (Pascanu et al., 2013).

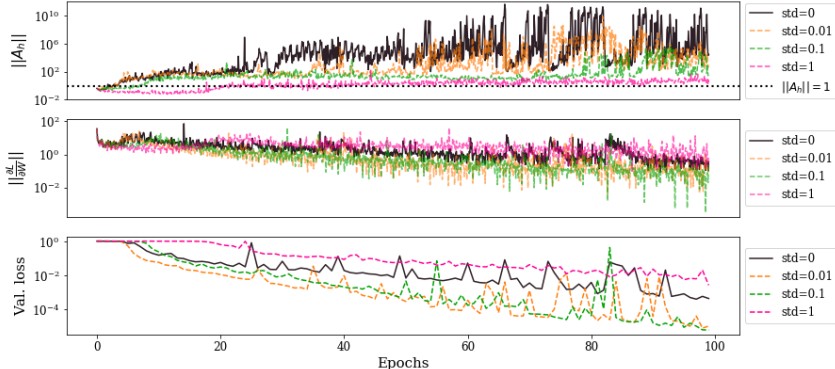

Figure 4: Visualization of the regularization effect of a random initial hidden state on the adding task ($T = 100$). (Top) Norm of $\mathcal{A}_h$ every 100 iterations; (Middle) norm of the gradient of the recurrent weight every 100 iterations; (Bottom) validation loss at every epoch. Each epoch contains 1000 iterations.

Our key realization is that the gradient of the RNN output with respect to the initial hidden state $\boldsymbol{h}^{(0)}$ features the term $\mathcal{A}_h$ from Theorem 3

$$\frac{\mathrm{d}\mathcal{L}}{\mathrm{d}\boldsymbol{h}^{(0)}} = \frac{\mathrm{d}\mathcal{L}}{\mathrm{d}\boldsymbol{z}} \boldsymbol{W} \left( \prod_{s=T}^{1} \boldsymbol{A}_{\sigma}^{(s)} \boldsymbol{W}_r \right) = \frac{\mathrm{d}\mathcal{L}}{\mathrm{d}\boldsymbol{z}} \boldsymbol{W} \mathcal{A}_h \,. \tag{11}$$

Of all the terms in (10), this one involves the most matrix products and hence is the most erratic. Fortunately, Theorem 3 instructs us that introducing randomness into the initial hidden state effects a regularization on $\mathcal{A}_h$ and hence tamps down the gradient before it can explode. An interesting direction for future work is extending this analysis to every term in (10).

**Experiments.** We now report on the results of a number of experiments that indicate the significant performance gains that can be obtained using a random initial hidden state of properly chosen standard deviation $\sigma_\epsilon$. Unless otherwise mentioned, in all experiments we use ReLU RNNs with 128-dimensional hidden states and with the recurrent weight matrix $\boldsymbol{W}_r^{(\ell)}$ initialized as an identity matrix (Le et al., 2015; Talathi & Vartak, 2016). We summarize the experimental results; experimental details and additional results are available in Appendices C and F.

*Visualizing the Regularizing Effect of a Random Initial Hidden State.* We first consider a simulated task of adding 2 sequences of length 100. This is a ternary classification problem with input $\boldsymbol{X} \in \mathbb{R}^{2 \times T}$ and target $y \in \{0, 1, 2\}, y = \sum_i \mathbb{1}_{\boldsymbol{X}_{2i}=1} \boldsymbol{X}_{1i}$. The first row of $\boldsymbol{X}$ contains randomly chosen 0's and 1's; the second row of $\boldsymbol{X}$ contains 1's at 2 randomly chosen indices and 0's everywhere else. Prior work treats this task as a regression task (Arjovsky et al., 2016); our regression results are provided in Appendix F.1.

In Figure 4, we visualize the norm of $\mathcal{A}_h$, the norm of the recurrent weight gradient $\|\frac{\mathrm{d}\mathcal{L}}{\mathrm{d}\boldsymbol{W}_r}\|$, and the validation loss against training epochs for various random initial state standard deviations. The top two plots clearly demonstrate the effect of the random initial hidden state in regularizing both $\mathcal{A}_h$ and the norm of the recurrent weight gradient, since larger $\sigma_\epsilon$ reduces the magnitudes of both $\mathcal{A}_h$ and $\|\frac{\mathrm{d}\mathcal{L}}{\mathrm{d}\boldsymbol{W}_r}\|$. Notably, the reduced magnitude of the gradient term $\|\frac{\mathrm{d}\mathcal{L}}{\mathrm{d}\boldsymbol{W}_r}\|$ empirically demonstrates the mollification of the exploding gradient problem. The bottom plot shows that setting $\sigma_\epsilon$ too large can negatively impact learning. This can be explained as having too much regularization effect. This suggests the question of choosing the best value of $\sigma_\epsilon$ in practice, which we now investigate.

*Choosing the Standard Deviation of the Random Initial Hidden State.* We examine the effect on performance of different random initial state standard deviations $\sigma_\epsilon$ in RNNs using RMSprop and SGD with varying learning rates. We perform experiments on the MNIST dataset with each image flattened to a length 784 sequence (recall Section 4). The full experimental results are included in Appendix F.2; here, we report two interesting findings. First, for both optimizers, using a random initial hidden state permits the use of higher learning rates that would lead to an exploding gradient when training without a random initial hidden state. Second, RMSprop is less sensitive to the choice of $\sigma_\epsilon$ than SGD and achieves favorable accuracy even when $\sigma_\epsilon$ is very large (e.g., $\sigma_\epsilon = 5$). This might be due to the gradient smoothing that RMSprop performs during optimization. We therefore recommend the use of RMSprop with a random initial hidden state to improve model performance.

Table 1: Classification accuracies on the (permuted) MNIST and SST-2 test sets for various models. A random initial hidden state elevates simple RNNs from also-rans to strong competitors of complex, state-of-the-art models.

| Model | Dataset | | |
|---|---|---|---|
| | MNIST | permuted MNIST | SST-2 |
| RNN, 1 layer, zero initial hidden state | 0.970 | 0.891 | 0.871 |
| RNN, 1 layer, random initial hidden state | 0.981 | 0.922 | 0.873 |
| | $(\sigma_\epsilon = 0.1)$ | $(\sigma_\epsilon = 0.01)$ | $(\sigma_\epsilon=0.1)$ |
| RNN, 2 layers, zero initial hidden state | 0.969 | 0.873 | 0.884 |
| RNN, 2 layers, random initial hidden state | **0.987** | 0.927 | 0.888 |
| | $(\sigma_\epsilon = 0.5)$ | $(\sigma_\epsilon = 0.005)$ | $(\sigma_\epsilon=0.005)$ |
| GRU | 0.986 | 0.888 | 0.881 |
| LSTM | 0.978 | 0.913 | 0.849 |
| uRNN (Arjovsky et al., 2016) | 0.951 | 0.914 | – |
| scoRNN (Helfrich et al., 2018) | 0.985 | **0.966** | – |
| C-LSTM (Zhou et al., 2015) | – | – | 0.878 |
| Tree-LSTM (Tai et al., 2015) | – | – | 0.88 |
| Bi-LSTM+SWN-Lex (Teng et al., 2016) | – | – | **0.892** |

We used RMSprop to train ReLU RNNs of one and two layers with and without random initial hidden state on the MNIST, permuted MNIST[4] and SST-2 datasets. Table 1 shows the classification accuracies of these models as well as a few state-of-the-art results using complicated models. It is surprising that a random initial hidden state elevates the performance of a simple ReLU RNN to near state-of-the-art performance.

*Random Initial Hidden State in Complex RNN Models.* Inspired by the results of the previous experiment, we integrated a random initial hidden state into some more complex RNN models. We first evaluate a one-layer gated recurrent unit (GRU) on the MNIST and permuted MNIST datasets, with a random and zero initial hidden state. Although the performance gains are not quite as impressive as those for ReLU RNNs, our results for GRUs still show worthwhile accuract improvements, from $0.986$ to $0.987$ for MNIST and from $0.888$ to $0.904$ for permuted MNIST.

We continue our experiments with a more complex, convolutional-recurrent model composed of 4 convolution layers followed by 2 GRU layers (Cakir et al., 2017) and the Bird Audio Detection Challenge dataset.[5] This binary classification problem aims to detect whether or not an audio recording contains bird songs; see Appendix C for the details. We use the area under the ROC curve (AUC) as the evaluation metric, since the dataset is highly imbalanced. Simply switching from a zero to a random initial hidden state provides a significant boost in the AUC: from 90.5% to 93.4%. These encouraging preliminary results suggest that, while more theoretical and empirical investigations are needed, a random initial hidden state can also boost the performance of complicated RNN models that are not piecewise affine and convex.

## 7 CONCLUSIONS AND FUTURE WORK

We have developed and explored a novel perspective of RNNs in terms of max-affine spline operators (MASOs). RNNs with piecewise affine and convex nonlinearities are piecewise affine spline operators with a simple, elegant analytical form. The connections to input space partitioning (vector quantization) and matched filtering followed immediately. The spline viewpoint also suggested that the typical zero initial hidden state be replaced with a random one that mollifies the exploding gradient problem and improves generalization performance.

There remain abundant promising research directions. First, we can extend the MASO RNN framework following (Balestriero & Baraniuk, 2018b) to cover more general networks like gated RNNs (e.g, GRUs, LSTMs) that employ the sigmoid nonlinearity, which is neither piecewise affine nor convex. Second, we can apply recent random matrix theory results (Martin & Mahoney, 2018) to the affine parameter $A_{RNN}$ (e.g., the change of the distribution of its singular values during training) to understand RNN training dynamics.

---

[4]We apply a fixed permutation to all MNIST images.

[5]The leaderboard of benchmarks can be found at https://goo.gl/TyaFrd.

## ACKNOWLEDGEMENTS

We thank the anonymous reviewers for their constructive feedback. This work was partially supported by NSF grants IIS-17-30574 and IIS-18-38177, AFOSR grant FA9550-18-1-0478, ARO grant W911NF-15-1-0316, ONR grants N00014-17-1-2551 and N00014-18-12571, DARPA grant G001534-7500, and DOD Vannevar Bush Faculty Fellowship (NSSEFF) grant N00014-18-1-2047.

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

## A  NOTATION

| | |
|---|---|
| $L, \ell$ | Total number of layers: $L \geq 0$; Index of the layer in an RNN: $\ell \in \{0, \cdots, L\}$ |
| $T, t$ | Total number of time steps: $T \geq 0$; Index of time steps of an RNN: $t \in \{0, \cdots, T\}$ |
| $C, c$ | Total number of output classes: $C > 0$; Index of the output class: $c \in \{1, \cdots, C\}$ |
| $N, n$ | total number of examples: $N \geq 1$;
Index of example in a dataset: $n \in \{1, \cdots, N\}$ |
| $R^{(\ell)}$ | Index of the partition region induced by the piecewise nonlinearity at layer $l$ |
| $D^{(\ell)}$ | Dimension of input to the RNN at layer $\ell$ |
| $K, k$ | Total number of output dimensions of a MASO, $K \geq 0$; Index of MASO output dimension, $k \in \{1, \cdots, K\}$ |
| $Q$ | The partition region selection matrix |
| $\boldsymbol{x}^{(t)}$ | $t^{\text{th}}$ time step of a discrete time-serie, $\boldsymbol{x}^{(t)} \in \mathbb{R}^{D^{(0)}}$ |
| $\boldsymbol{x}$ | Concatenation of the whole length $T$ time-serie: $\boldsymbol{x} = \left[ \boldsymbol{x}^{(1)^\top}, \cdots, \boldsymbol{x}^{(T)^\top} \right]^\top, \boldsymbol{x} \in \mathbb{R}^{D^{(0)}T}$ |
| $\boldsymbol{X}$ | A dataset of $N$ time-series: $\boldsymbol{X} = \{\boldsymbol{x}_n\}_1^N$ |
| $\widehat{y}(\boldsymbol{x})$ | Output/prediction associated with input $\boldsymbol{x}$ |
| $y_n$ | True label (target variable) associated with the $n$th time-serie example $\boldsymbol{x}_n$.
For classification $y_n \in \{1, \ldots, C\}$, $C > 1$; For regression $y_n \in \mathbb{R}^C, C \geq 1$ |
| $\boldsymbol{h}^{(\ell,t)}$ | Output of an RNN cell at layer $\ell$ and time step $t$;
Alternatively, input to an RNN cell at layer $\ell + 1$ and time step $t - 1$ |
| $\boldsymbol{h}^{(\ell)}$ | Concatenation of hidden state $\boldsymbol{h}^{(\ell,t)}$ of all time steps at layer $\ell$: $\boldsymbol{h}^{(\ell)} = \left[ \boldsymbol{h}^{(\ell,1)^\top}, \cdots, \boldsymbol{x}^{(\ell,T)^\top} \right]^\top, \boldsymbol{h}^{(\ell)} \in \mathbb{R}^{D^{(\ell)}T}$ |
| $\boldsymbol{z}^{(\ell,t)}$ | Concatenated input to an RNN cell at layer $\ell$ and time step $t$: $\boldsymbol{z}^{(\ell,t)} = \left[ \boldsymbol{h}^{(\ell-1,t)^\top}, \boldsymbol{h}^{(\ell,t-1)^\top} \right]^\top, \boldsymbol{z}^{(\ell,t)} \in \mathbb{R}^{2D^{(\ell)}}$ |
| $\boldsymbol{W}_r^{(\ell)}$ | $\ell$th layer RNN weight associated with the input $\boldsymbol{h}^{(\ell,t-1)}$ from the previous time step: $\boldsymbol{W}_r^{(\ell)} \in \mathbb{R}^{D^{(\ell)} \times D^{(\ell)}}$ |
| $\boldsymbol{W}^{(\ell)}$ | $\ell$th layer RNN weight associated with the input $\boldsymbol{h}^{(\ell-1,t)}$ from the previous layer: $\boldsymbol{W}^{(\ell)} \in \mathbb{R}^{D^{(\ell)} \times D^{(\ell-1)}}$ |
| $\boldsymbol{W}$ | Weight of the last fully connected layer: $\boldsymbol{W} \in \mathbb{R}^{C \times D^{(L)}}$ |
| $\boldsymbol{b}^{(\ell)}$ | $\ell$th layer RNN bias: $\boldsymbol{b}^{(\ell)} \in \mathbb{R}^{D^{(\ell)}}$ |
| $\boldsymbol{b}$ | Bias of the last fully connected layer: $\boldsymbol{b} \in \mathbb{R}^C$ |
| $\sigma(\cdot)$ | Pointwise nonlinearity in an RNN (assumed to be piecewise affine and convex in this paper) |
| $\sigma_{\boldsymbol{\epsilon}}$ | Standard deviation of noise injected into the initial hidden state $\boldsymbol{h}^{(\ell,0)} \forall \ell$ |
| $A_\sigma^{(\ell,t)}$ | MASO formula of the RNN activation $\sigma(\cdot)$ at layer $\ell$ and time step $t$: $A_\sigma \in \mathbb{R}^{D^\ell \times D^{(\ell)}}$ |
| $A^{(\ell,t)}, B^{(\ell,t)}$ | MASO parameters of an RNN at layer $\ell$ and time step $t$:
$A^{(\ell,t)} \in \mathbb{R}^{D^{(\ell)} \times R^{(\ell)} \times D^{(\ell-1)}}, B^{(\ell,t)} \in \mathbb{R}^{D^{(\ell)} \times R^{(\ell)}}$ |

## B  DATASETS AND PREPROCESSING STEPS

Below we describe the datasets and explain the preprocessing steps for each dataset.

**MNIST**. The dataset[6] consists of 60k images in the training set and 10k images in the test set. We randomly select 10k images from the training set as validation set. We flatten each image to a 1-

---

[6] http://yann.lecun.com/exdb/mnist/

12

dimensional vector of size 784. Each image is also centered and normalized with mean of 0.1307 and standard deviation of 0.3081 (PyTorch default values).

**permuted MNIST**. We apply a fixed permutation to all images in the MNIST dataset to obtain the permuted MNIST dataset.

**SST-2**. The dataset[7] consists of 6920, 872, 1821 sentences in the training, validation and test set, respectively. Total number of vocabulary is 17539, and average sentence length is 19.67. Each sentence is minimally processed into sequences of words and use a fixed-dimensional and trainable vector to represent each word. We initialize these vectors either randomly or using GloVe (Pennington et al., 2014). Due to the small size of the dataset, the phrases in each sentence that have semantic labels are also used as part of the training set in addition to the whole sentence during training. Dropout of 0.5 is applied to all experiments. Phrases are not used during validation and testing, i.e., we always use entire sentences during validation and testing.

**Bird Audio Dataset**. The dataset[8] consists of $7,000$ field recording signals of 10 seconds sampled at 44 kHz from the Freesound Stowell & Plumbley (2014) audio archive representing slightly less than 20 hours of audio signals. The audio waveforms are extracted from diverse scenes such as city, nature, train, voice, water, etc., some of which include bird sounds. The labels regarding the bird detection task can be found in the file `freefield1010`. Performance is measured by Area Under Curve (AUC) due to the unbalanced distribution of the classes. We preprocess every audio clip by first using short-time Fourier transform (STFT) with 40ms and 50% overlapping Hamming window to obtain audio spectrum and then by extracting 40 log mel-band energy features. After preprocessing, each input is of dimension $D = 96$ and $T = 999$.

## C  EXPERIMENTAL SETUP

Experiment setup for various datasets is summarized in Table 2. Some of the experiments do not appear in the main text but in the appendix; we include setup for those experiments as well. A setting common to all experiments is that we use learning rate scheduler so that when validation loss plateaus for 5 consecutive epochs, we reduce the current learning rate by a factor of 0.7.

**Setup of the experiments on influence of various standard deviations in random initial hidden state under different settings**. We use $\sigma_\epsilon$ chosen in $\{0.001, 0.01, 0.1, 1, 5\}$ and learning rates in $\{1 \times 10^{-5}, 1 \times 10^{-4}, 1.5 \times 10^{-4}, 2 \times 10^{-4}\}$ for RMSprop and $\{1 \times 10^{-7}, 1 \times 10^{-6}, 1.25 \times 10^{-6}, 1.5 \times 10^{-6}\}$ plain SGD.

**Setup of input space partitioning experiments**. For the results in the main text, we use t-SNE visualization (van der Maaten & Hinton, 2008) with 2 dimensions and the default settings from the python `sklearn` package. Visualization is performed on the whole 10k test set images. For finding the nearest neighbors of examples in the SST-2 dataset, since the examples are of varying lengths, we constrain the distance comparison to within +/-10 words of the target sentence. When the sentence lengths are not the same, we simply pad the shorter ones to the longest one, then process it with RNN and finally calculate the distance as the $\ell_2$ distance of the partition codes (i.e., concatenation of all hidden states) that RNN computes. We justify the comparison between examples of different lengths using padding by noting that batching examples and padding the examples to the longest example within a batch has been a common practice in modern natural language processing tasks.

**Setup of exploratory experiments**. We experimented with one-layer GRU with 128 hidden unites for MNIST and permuted MNIST datasets. We use RMSprop optimizer with an initial learning rate of 0.001. We experimented with various standard deviations in random initial hidden state including $\{0.01, 0.05, 0.1, 0.5, 1, 5\}$. The optimal standard deviations that produce the results in the main text are $\sigma_\epsilon = \{0.01, 0.05, 0.01\}$, for MNIST, permuted MNIST and bird detection datasets, respectively.

## D  ADDITIONAL INPUT SPACE PARTITION VISUALIZATIONS

We provide ample additional visualizations to demonstrate the partition codes that an RNN computes on its input sequences. Here, the results are focused more on the properties of the final partition

---

[7] https://nlp.stanford.edu/sentiment/index.html

[8] http://machine-listening.eecs.qmul.ac.uk/bird-audio-detection-challenge/

Table 2: Various experiment setup. Curly brackets indicate that we attempted more than one value for this experiment.

| Settings | Dataset | | | | |
|---|---|---|---|---|---|
| | Add task | MNIST | Permuted MNIST | SST-2 | Bird detection |
| Type of RNN | ReLU RNN | ReLU RNN | ReLU RNN | ReLU RNN | GRU |
| #Layers | {1, 2} | {1, 2} | {1, 2} | {1, 2} | 2 |
| Input size | 2 | 1 | 1 | 128 | 96 |
| Hidden size | 128 | 128 | 128 | 300 | 256 |
| Output size | 3 | 3 | 10 | 2 | 2 |
| Initial Learning Rate | 1e-4 | 1e-4 | 1e-4 | 1e-4 | 1e-4 |
| Optimizer | RMSprop | {RMSprop, SGD} | RMSprop | Adam | Adam |
| Batch size | 50 | 64 | 64 | 64 | 64 |
| Epochs | 100 | 200 | 200 | 100 | 50 |

Figure 5: Visualization of partition codes for pixel-by-pixel (i.e., flattened to a 1-dimensional, length 784 vector) MNIST dataset using a trained ReLU RNN (one layer, 128-dimensional hidden state). Here, we visualize the nearest 5 and farthest 5 images of one selected image from each class. The distance is computed using the partition codes of the images. Leftmost column is the original image; the middle 5 images are the 5 nearest neighbors; the rightmost 5 images are the farthest neighbors.

codes computed after the RNN processes the entire input sequence rather than part of the input sequence. Several additional sets of experimental results are shown; the first three on MNIST and the last one on SST-2.

First, we visualize the nearest and farthest neighbors of several MNIST digits in Figure 5. Distance is calculated using the partition codes of the images. The left column is the original image; The next five columns are the five nearest neighbors to the original image; The last five columns are five farthest neighbors. This figure shows that partition codes the images are well clustered.

Second, we show the two dimensional projection using t-SNE of the raw pixel and VQ representations of each data points in the MNIST dataset and visualize them in Figure 6. We clearly see a more distinct clustering using VQ representation of the data than using the raw pixel representation. This comparison demonstrate the ability of the RNN to extract useful information from the raw representation of the data in the form of VQ.

Third, we perform a KNN classification with $k \in \{1, 2, 5, 10\}$ using 1) the RNN computed partition codes of the inputs and 2) raw pixel data representation the MNIST test set to illustrate that the data reparametrized by the RNN has better clustering property than the original data representations. We use 80% of the test set to train the classifier and the rest for testing. The results are reported in Table 3. We see that the classification accuracies when using RNN computed partition codes of the inputs are significantly higher than those when using raw pixel representations. This result again shows the superior quality of the input space partitioning that RNN produces, and may suggest a new way to improve classification accuracy by just using the reparametrized data with a KNN classifier.

Finally, we visualize the 5 nearest and 5 farthest neighbors of a selected sentence from the SST-2 dataset to demonstrate that the partitioning effect on dataset of another modality. Again, the distances are computed using the partition codes of the inputs. The results are shown in Figure 7.

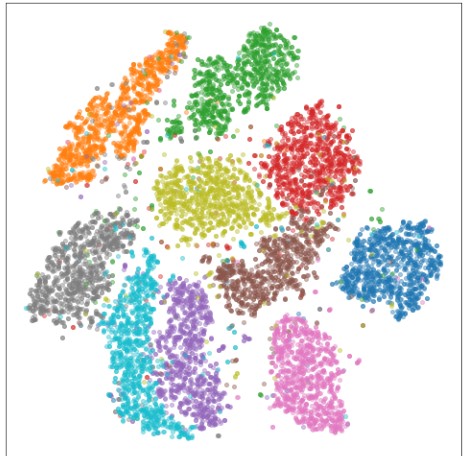 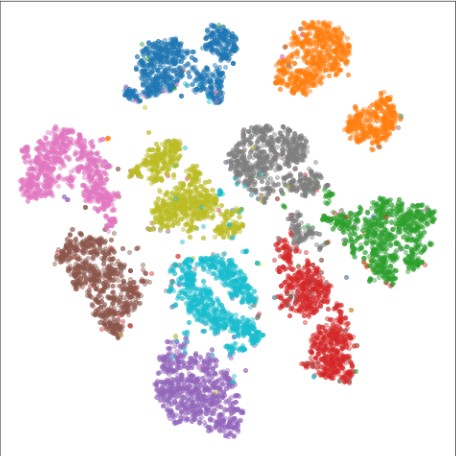

Figure 6: t-SNE visualization of MNIST test set images using raw pixel representation (**left**) and RNN VQ representation (**right**). We see more distinct clusters in the t-SNE plot using RNN VQ representation of images than the raw pixel representation, implying the useful information that RNN extracts in the form of VQ.

Table 3: K-nearest neighbor classification accuracies using data reparametrized by RNN compared to those using raw pixel data. We can see that classification accuracies using RNN reparametrized data are much higher than those using raw pixel data for all $k$'s.

| k | MNIST, raw pixels | MNIST, VQ |
|---|---|---|
| 1 | 0.950 | **0.977** |
| 2 | 0.936 | **0.974** |
| 5 | 0.951 | **0.977** |
| 10 | 0.939 | **0.975** |

We can see that all sentences that are nearest neighbors are of similar sentiment to the target sentence, whereas all sentences that are farthest neighbors are of the opposite sentiment.

## E    ADDITIONAL TEMPLATE VISUALIZATIONS

We provide here more templates on images and texts in Figures 8 and 9. Notice here that, although visually the templates may look similar or meaningless, they nevertheless have meaningful inner product with the input. The class index of the template that produces the largest inner product with the input is typically the correct class, as can be seen in the two figures.

## F    ADDITIONAL EXPERIMENTAL RESULTS FOR RANDOM INITIAL HIDDEN STATE

### F.1    REGULARIZATION EFFECT FOR REGRESSION PROBLEM

We present the regularization effect on adding task formulated as a regression problem, following setup in Arjovsky et al. (2016). Result is shown in Figure 10. We see regularization effect similar to that presented in Figure 4, which demonstrates that the regularization effect does indeed happens for both classification and regression problems, as Thm. 3 suggests.

### F.2    CHOOSING STANDARD DEVIATION IN RANDOM INITIAL HIDDEN STATE

Table 4 shows the classification accuracies under various settings. The discussion of the results is in Section 6.

| Original text |
| --- |
| It is a film that will have people walking out halfway through , will encourage others to stand up and applaud , and will , undoubtedly , leave both camps engaged in a ferocious debate for years to come . (**+**) |

| Nearest 5 neighbors | Farthest 5 neighbors |
| --- | --- |
| Well-written , nicely acted and beautifully shot and scored , the film works on several levels , openly questioning social mores while ensnaring the audience with its emotional pull . (**+**, 22.00) | Marries the amateurishness of The Blair Witch Project with the illogic of Series 7 : The Contenders to create a completely crass and forgettable movie . (**-**, 37.60) |
| A stunning piece of visual poetry that will , hopefully , be remembered as one of the most important stories to be told in Australia 's film history . (**+**, 22.23) | K-19 may not hold a lot of water as a submarine epic , but it holds even less when it turns into an elegiacally soggy Saving Private Ryanovich . (**-**, 37.42) |
| Cute , funny , heartwarming digitally animated feature film with plenty of slapstick humor for the kids , lots of in-jokes for the adults and heart enough for everyone . (**+**, 22.61) | This is a great subject for a movie , but Hollywood has squandered the opportunity , using it as a prop for warmed-over melodrama and the kind of choreographed mayhem that director John Woo has built his career on . (**-**, 37.24) |
| Though it is by no means his best work , Laissez-Passer is a distinguished and distinctive effort by a bona-fide master , a fascinating film replete with rewards to be had by all willing to make the effort to reap them . (**+**, 22.78) | Flotsam in the sea of moviemaking , not big enough for us to worry about it causing significant harm and not smelly enough to bother despising . (**-**, 37.15) |
| An absorbing trip into the minds and motivations of people under stress as well as a keen , unsentimental look at variations on the theme of motherhood . (**+**, 22.89) | If you 're not a prepubescent girl , you 'll be laughing at Britney Spears ' movie-starring debut whenever it does n't have you impatiently squinting at your watch . (**-**, 36.98) |

Figure 7: Nearest and furthest neighbors of a postive movie review. The sentiment (+ or -) and the euclidean distance between the input and the neighbor vector quantizations are shown in parenthesis after each neighbor.

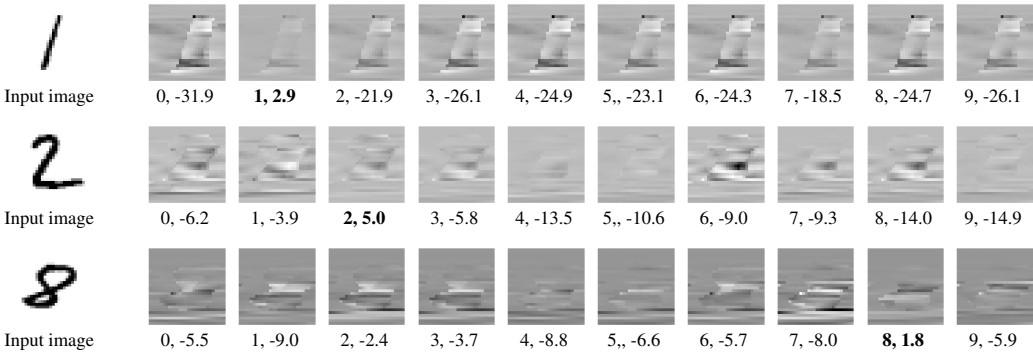

Figure 8: Templates of three selected MNIST images. The leftmost column are the original input image. The next ten images of each row are the ten templates of a particular input image corresponding to each class. For each template image, we show the class and the inner product of this template with the input. Text under the template of the true class of each input image is bolded.

# G  PROOFS

## G.1  PROOF OF THM. 1

To simplify notation, similar to the main text, in the proof here we drop the affine parameters' dependencies on the input, but keep in mind the input-dependency of these parameters.

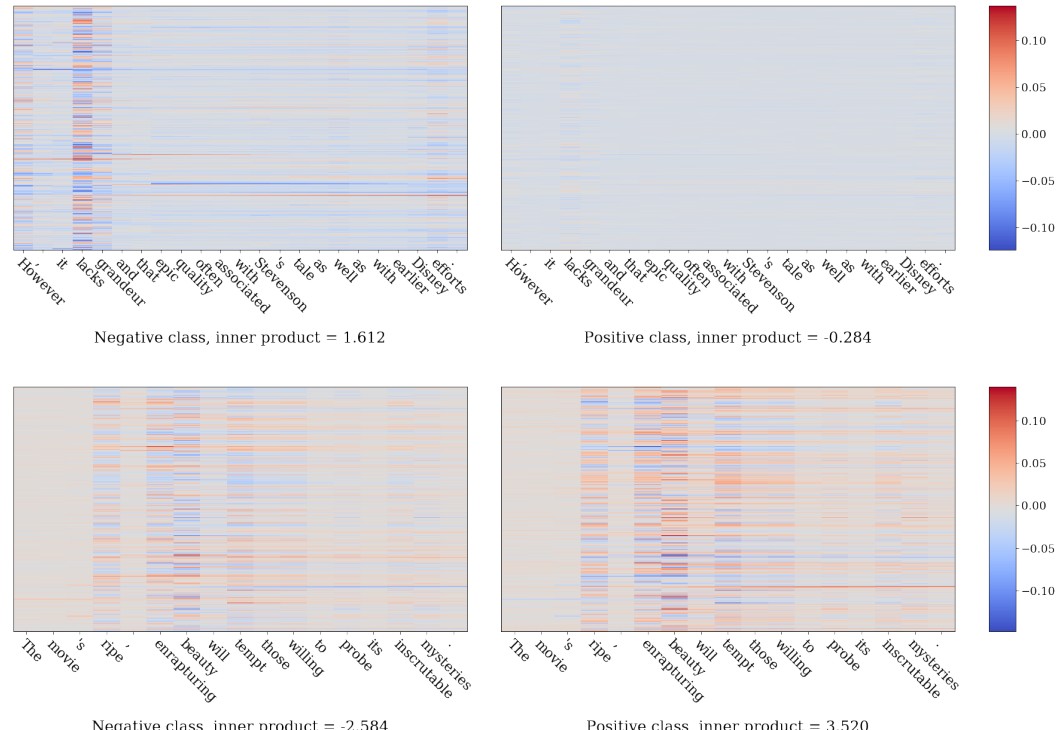

Figure 9: Additional template visualizations of an example from the SST-2 dataset. Each word in the sentence is marked as a tick label in the $x$ axis. The values of inner products are marked below each template. The template that has the bigger inner product is the true class of the sentence. We see that the template corresponding to the correct class produces a significantly bigger inner product with the input than other templates.

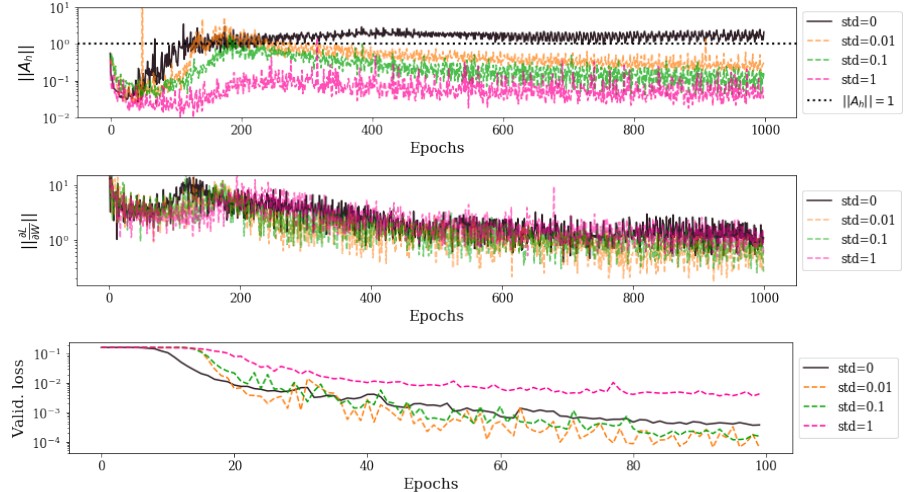

Figure 10: Various plots during training of add problem (T=100, regression). **Top**: norm of $\mathbf{A}_h$ at every 100 iterations; **Middle**: norm of gradient of recurrent weight at every 100 iterations; **Bottom**: validation loss at every epoch. Each epoch contains 1000 iterations.

We first derive the expression for a hidden state $\boldsymbol{h}^{(\ell,t)}$ at a given time step $t$ and layer $\ell$. Using Prop. 1, we start with unrolling the RNN cell of layer $\ell$ at time step $t$ for two time steps to $t-2$ by recursively applying (1) as follows:

Table 4: Classification accuracy for MNIST dataset under 2 different optimizers, various learning rates and different standard deviation $\sigma_\epsilon$ in the random initial hidden state. Results suggest RMSprop tolerates various choices of $\sigma_\epsilon$ while SGD works for smaller $\sigma_\epsilon$.

| | **RMSprop** | | | | **SGD** | | | |
|---|---|---|---|---|---|---|---|---|
| $\sigma_\epsilon$ | 1e-5 | 1e-4 | 1.5e-4 | 2e-4 | 1e-7 | 1e-6 | 1.25e-6 | 1.5e-6 |
| 0 | 0.960 | 0.973 | 0.114 | 0.114 | **0.837** | 0.879 | 0.870 | 0.098 |
| 0.001 | **0.963** | 0.974 | 0.978 | 0.970 | 0.835 | 0.895 | 0.913 | 0.875 |
| 0.01 | 0.962 | 0.980 | 0.978 | 0.976 | 0.834 | **0.898** | **0.922** | **0.918** |
| 0.1 | 0.955 | 0.976 | **0.981** | 0.976 | 0.803 | 0.833 | 0.913 | 0.908 |
| 1 | 0.956 | 0.977 | 0.980 | 0.976 | 0.520 | 0.640 | 0.901 | 0.098 |
| 5 | 0.952 | **0.981** | 0.973 | **0.981** | 0.471 | 0.098 | 0.098 | 0.098 |

$$\boldsymbol{h}^{(\ell,t)} = \sigma\left(\boldsymbol{W}^{(\ell)}\boldsymbol{h}^{(\ell-1,t)} + \boldsymbol{b}^{(\ell)} + \boldsymbol{W}_r^{(\ell)}\boldsymbol{h}^{(\ell,t-1)}\right) \tag{12}$$

$$= A_\sigma^{(\ell,t)}\boldsymbol{W}^{(\ell)}\boldsymbol{h}^{(\ell-1,t)} + A_\sigma^{(t)}\boldsymbol{b}^{(\ell)} + A_\sigma^{(t)}\boldsymbol{W}_r^{(\ell)}\boldsymbol{h}^{(\ell,t-1)} \tag{13}$$

$$= A_\sigma^{(\ell,t)}\boldsymbol{W}^{(\ell)}\boldsymbol{h}^{(\ell-1,t)} + A_\sigma^{(t)}\boldsymbol{b}^{(\ell)}$$
$$+ A_\sigma^{(\ell,t)}\boldsymbol{W}_r^{(\ell)}\left(A_\sigma^{(\ell,t-1)}\boldsymbol{W}^{(\ell)}\boldsymbol{h}^{(\ell-1,t-1)} + A_\sigma^{(\ell,t-1)}\boldsymbol{b}^{(\ell)} + A_\sigma^{(\ell,t-1)}\boldsymbol{W}_r^{(\ell)}\boldsymbol{h}^{(\ell,t-2)}\right) \tag{14}$$

$$= \left(A_\sigma^{(\ell,t)}\boldsymbol{W}^{(\ell)}\boldsymbol{h}^{(\ell-1,t)} + A_\sigma^{(\ell,t)}\boldsymbol{W}_r^{(\ell)}A_\sigma^{(\ell,t-1)}\boldsymbol{W}^{(\ell)}\boldsymbol{h}^{(\ell-1,t-1)}\right)$$
$$+ \left(A_\sigma^{(\ell,t)} + A_\sigma^{(\ell,t)}\boldsymbol{W}_r^{(\ell)}A_\sigma^{(\ell,t-1)}\right)\boldsymbol{b}^{(\ell)} + A_\sigma^{(\ell,t)}\boldsymbol{W}_r^{(\ell)}A_\sigma^{(\ell,t-1)}\boldsymbol{W}_r^{(\ell)}\boldsymbol{h}^{(\ell,t-2)}. \tag{15}$$

From (12) to (13), we use the result of Prop. 1. From (13) to (14), we expand the term that involves the hidden state $\boldsymbol{h}^{(\ell,t-1)}$ at the previous time step recursively using (13). From (14) to (15), we group terms and write them compactly.

Now define $\mathcal{A}_{s:t}^{(\ell,s)} := \prod_{k=t}^{s+1} A_\sigma^{(\ell,k)}\boldsymbol{W}_r^{(\ell)}$ for $s < t$ and $\mathcal{A}_{t:t}^{(\ell,t)} := \boldsymbol{I}$ where $\boldsymbol{I}$ is the identity matrix. Using this definition, we rewrite (15) and proceed with the unrolling to the initial time step as follows:

$$\boldsymbol{h}^{(\ell,t)} = \begin{pmatrix}\mathcal{A}_{t:t}^{(\ell)} & \mathcal{A}_{t-1:t}^{(\ell)}\end{pmatrix}\begin{pmatrix}A_\sigma^{(\ell,t)}\boldsymbol{W}^{(\ell)} \\ A_\sigma^{(\ell,t-1)}\boldsymbol{W}^{(\ell)}\end{pmatrix}\begin{pmatrix}\boldsymbol{h}^{(\ell-1,t)} \\ \boldsymbol{h}^{(\ell-1,t-1)}\end{pmatrix} + \sum_{s=t}^{t-1}\mathcal{A}_{s:t}^{(\ell)}B^{(\ell,s)} \tag{16}$$

$$\cdots$$

$$= \begin{pmatrix}\mathcal{A}_{t:t}^{(\ell)} & \cdots & \mathcal{A}_{1:t}^{(\ell)}\end{pmatrix}\begin{pmatrix}A_\sigma^{(\ell,t)}\boldsymbol{W}^{(\ell)} \\ \vdots \\ A_\sigma^{(\ell,1)}\boldsymbol{W}^{(\ell)}\end{pmatrix}\begin{pmatrix}\boldsymbol{h}^{(\ell-1,t)} \\ \vdots \\ \boldsymbol{h}^{(\ell-1,1)}\end{pmatrix} + \sum_{s=t}^{t-1}\mathcal{A}_{s:t}^{(\ell)}B^{(\ell,s)} + \mathcal{A}_{0:t}^{(\ell)}\boldsymbol{h}^{(\ell,0)}, \tag{17}$$

where $B^{(\ell,s)} = A_\sigma^{(\ell,s)}\boldsymbol{b}^{(\ell)}$ as defined in Prop. 1.

Repeat the above derivation for $t \in \{1, \cdots, T\}$ and stack $\boldsymbol{h}^{(\ell,t)}$ in decreasing time steps from top to bottom, we have:

$$
\begin{pmatrix} \boldsymbol{h}^{(\ell,T)} \\ \vdots \\ \boldsymbol{h}^{(\ell,1)} \end{pmatrix} = \begin{pmatrix} \mathcal{A}_{T:T}^{(\ell)} & \cdots & \mathcal{A}_{1:T}^{(\ell)} \\ \vdots & \ddots & \vdots \\ \boldsymbol{0} & \cdots & \mathcal{A}_{1:1}^{(\ell)} \end{pmatrix} \begin{pmatrix} A_\sigma^{(\ell,T)} \boldsymbol{W}^{(\ell)} & \cdots & \boldsymbol{0} \\ \vdots & \ddots & \vdots \\ \boldsymbol{0} & \cdots & A_\sigma^{(\ell,1)} \boldsymbol{W}^{(\ell)} \end{pmatrix} \begin{pmatrix} \boldsymbol{h}^{(\ell-1,T)} \\ \vdots \\ \boldsymbol{h}^{(\ell-1,1)} \end{pmatrix}
$$

$$
+ \begin{pmatrix} \sum_{t=T}^{1} \mathcal{A}_{t:T}^{(\ell)} B^{(\ell,t)} + \mathcal{A}_{0:T}^{(\ell)} \boldsymbol{h}^{(\ell,0)} \\ \vdots \\ \mathcal{A}_{1:1}^{(\ell)} B^{(\ell,t)} + \mathcal{A}_{0:1}^{(\ell)} \boldsymbol{h}^{(\ell,0)} \end{pmatrix} = A_{\mathrm{RNN}}^{(\ell)} \boldsymbol{h}^{(\ell-1)} + B_{\mathrm{RNN}}^{(\ell)}, \quad (18)
$$

where

$$
A_{\mathrm{RNN}}^{(\ell)} = \begin{pmatrix} \mathcal{A}_{T:T}^{(\ell)} & \cdots & \mathcal{A}_{1:T}^{(\ell)} \\ \vdots & \ddots & \vdots \\ \boldsymbol{0} & \cdots & \mathcal{A}_{1:1}^{(\ell)} \end{pmatrix}, \ B_{\mathrm{RNN}}^{(\ell)} = \begin{pmatrix} \sum_{t=T}^{1} \mathcal{A}_{t:T}^{(\ell)} B^{(\ell,t)} + \mathcal{A}_{0:T}^{(\ell)} \boldsymbol{h}^{(\ell,0)} \\ \vdots \\ \mathcal{A}_{1:1}^{(\ell)} B^{(\ell,t)} + \mathcal{A}_{0:1}^{(\ell)} \boldsymbol{h}^{(\ell,0)} \end{pmatrix}, \ \boldsymbol{h}^{(\ell-1)} = \begin{pmatrix} \boldsymbol{h}^{(\ell-1,T)} \\ \vdots \\ \boldsymbol{h}^{(\ell-1,1)} \end{pmatrix},
$$

which concludes the proof.

Thm. 2 follows from the recursive application of the above arguments for each layer $\ell \in \{1, \cdots, L\}$.

### G.2 PROOF OF THM. 3

We prove for the case of multi-class classification problem with softmax output. The proof for the case of regression problems easily follows.

Let $\boldsymbol{a}_i$ be the $i^{\text{th}}$ row of the input-dependent affine parameter $\mathcal{A}_h$ where $\mathcal{A}_h := \mathcal{A}_{1:T} = \prod_{s=T}^{1} A_\sigma^{(\ell,s)} \boldsymbol{W}_r^{(\ell)}$ (recall Section 6), $\boldsymbol{x}_n = [\boldsymbol{x}_n^{(1)^\top}, \cdots, \boldsymbol{x}_n^{(T)^\top}]^\top$ be the concatenation of the $n^{\text{th}}$ input sequence of length $T$ and $c$ be the index of the correct class. We assume the amplitude of random initial hidden state is small so that the input-dependent affine parameter $\mathcal{A}_h$, which also depends on $\boldsymbol{h}^{(0)}$, does not change when using random $\boldsymbol{h}^{(0)}$. Also, let $\boldsymbol{z}_n = f_{\mathrm{RNN}}(\boldsymbol{x}_n, \boldsymbol{h}^{(0)})$ be the overall RNN computation that represents (9).

We first rewrite the cross entropy loss with random initial hidden state $\widetilde{\mathcal{L}}_{\mathrm{CE}} = \mathcal{L}_{\mathrm{CE}} \left( \mathrm{softmax} \left( f_{\mathrm{RNN}} \left( \boldsymbol{x}_n, \boldsymbol{h}^{(0)} + \boldsymbol{\epsilon} \right) \right) \right)$ as follows:

$$
\widetilde{\mathcal{L}}_{\mathrm{CE}} = \frac{1}{N} \sum_{n=1}^{N} -\log \left( \mathrm{softmax} \left( f_{\mathrm{RNN}} \left( \boldsymbol{x}_n^{(1:T)}, \boldsymbol{h}^{(0)} + \boldsymbol{\epsilon} \right) \right) \right)
$$

$$
= \frac{1}{N} \sum_{n=1}^{N} -\log \left( \frac{\exp(z_{nc} + \boldsymbol{a}_c \boldsymbol{\epsilon})}{\sum_{j=1}^{C} \exp(z_{nj} + \boldsymbol{a}_j \boldsymbol{\epsilon})} \right)
$$

$$
= \frac{1}{N} \sum_{n=1}^{N} \left\{ -z_{nc} - \boldsymbol{a}_c \boldsymbol{\epsilon} + \log \left( \sum_{j=1}^{C} \exp(z_{nj} + \boldsymbol{a}_j \boldsymbol{\epsilon}) \right) \right\}. \quad (19)
$$

Taking expectation of the $\widetilde{\mathcal{L}}$ with respect to the distribution of the random Gaussian vector that the initial hidden state is set to, we have

$$
\mathbb{E}[\widetilde{\mathcal{L}}_{\mathrm{CE}}] = \mathcal{L}_{\mathrm{CE}} + \mathcal{R}, \quad (20)
$$

where

$$
\mathcal{R} = \frac{1}{N} \sum_{n=1}^{N} \left\{ \mathbb{E} \left[ \log \left( \sum_{j=1}^{C} \exp(z_{nj} + \boldsymbol{a}_j \boldsymbol{\epsilon}) \right) \right] - \log \left( \sum_{j=1}^{C} \exp(z_{nj}) \right) \right\}. \quad (21)
$$

We note that similar forms of (21) have been previously derived by Wager et al. (2013).

We now simplify (21) using second order Taylor expansion on $\boldsymbol{h}^{(0)}$ of the summation inside the log function. Define function $u(\boldsymbol{x}_n^{(1:T)}, \boldsymbol{h}^{(0)}) := \log(\sum_j \exp(z_{nj})) = \log(\sum_j \exp(f(\boldsymbol{x}_n^{(1:T)}, \boldsymbol{h}^{(0)})))$. Then, we can approximate (21) as follows:

$$
\begin{aligned}
\mathcal{R} &\approx \frac{1}{N} \sum_{n=1}^{N} \left\{ \mathbb{E}\left[ u(\boldsymbol{x}_n, \boldsymbol{h}^{(0)}) + \frac{du(\boldsymbol{x}_n, \boldsymbol{h}^{(0)})}{d\boldsymbol{h}^{(0)}}\boldsymbol{\epsilon} + \frac{1}{2}\boldsymbol{\epsilon}^\top \frac{du(\boldsymbol{x}_n, \boldsymbol{h}^{(0)})^2}{d^2\boldsymbol{h}^{(0)}}\boldsymbol{\epsilon} \right] - u(\boldsymbol{x}_n, \boldsymbol{h}^{(0)}) \right\} \\
&= \frac{1}{N} \sum_{n=1}^{N} \frac{1}{2}\mathbb{E}\left[ \mathrm{Tr}\left( \boldsymbol{\epsilon}^\top \frac{du(\boldsymbol{x}_n, \boldsymbol{h}^{(0)})^2}{d^2\boldsymbol{h}^{(0)}}\boldsymbol{\epsilon} \right) \right] \\
&= \frac{1}{N} \sum_{n=1}^{N} \frac{1}{2}\mathrm{Tr}\left( \frac{du(\boldsymbol{x}_n, \boldsymbol{h}^{(0)})^2}{d^2\boldsymbol{h}^{(0)}}\mathbb{E}\left[ \boldsymbol{\epsilon}\boldsymbol{\epsilon}^\top \right] \right) \\
&= \frac{1}{N} \sum_{n=1}^{N} \frac{\sigma_{\boldsymbol{\epsilon}}^2}{2}\mathrm{Tr}\left( \frac{du(\boldsymbol{x}_n, \boldsymbol{h}^{(0)})^2}{d^2\boldsymbol{h}^{(0)}} \right),
\end{aligned}
\tag{22}
$$

where $\frac{du(\boldsymbol{x}_n, \boldsymbol{h}^{(0)})^2}{d^2\boldsymbol{h}^{(0)}}$ is the Hessian matrix:

$$
\begin{aligned}
\left[ \frac{du(\boldsymbol{x}_n, \boldsymbol{h}^{(0)})^2}{d^2\boldsymbol{h}^{(0)}} \right]_{il} &= \frac{du(\boldsymbol{x}_n, \boldsymbol{h}^{(0)})^2}{d\boldsymbol{h}_i^{(0)}d\boldsymbol{h}_l^{(0)}} \\
&= \frac{d}{d\boldsymbol{h}_l^{(0)}} \underbrace{\frac{\exp(z_{ni})}{\sum_{j=1}^{C}\exp(z_{nj})}}_{\widehat{y}_{ni}} \boldsymbol{a}_i \\
&= \frac{d\widehat{y}_{ni}}{dz_{nl}}\frac{dz_{nl}}{\boldsymbol{h}_l^{(0)}}\boldsymbol{a}_i \\
&= \widehat{y}_{ni}(\mathbb{1}_{i=l} - \widehat{y}_{nl})\langle \boldsymbol{a}_i, \boldsymbol{a}_l \rangle.
\end{aligned}
$$

Then, we can write the trace term in (22) as follows:

$$
\mathrm{Tr}\left( \frac{du(\boldsymbol{x}_n, \boldsymbol{h}^{(0)})^2}{d^2\boldsymbol{h}^{(0)}} \right) = \left\| \mathrm{diag}\left( \left[ \frac{dy_{ni}}{dz_{nl}} \right]_{i=l} \right)\mathcal{A}_h \right\|^2.
$$

As a result, using the above approximations, we can rewrite the loss with random initial state in (19) as:

$$
\widetilde{\mathcal{L}}_{\mathrm{CE}} = \mathcal{L}_{\mathrm{CE}} + \frac{\sigma_{\boldsymbol{\epsilon}}^2}{2N}\sum_{n=1}^{N} \left\| \mathrm{diag}\left( \left[ \frac{dy_{ni}}{dz_{nl}} \right]_{i=l} \right)\mathcal{A}_h \right\|^2.
\tag{23}
$$

We see that this regularizer term does not dependent on the correct class index $c$ of each data points.

## H   PRIOR WORK ON THE EXPLODING GRADIENT IN RNNS

The problem of exploding gradients has been widely studied from different perspectives. First approaches have attempted to directly control the amplitude of the gradient through gradient clipping (Pascanu et al., 2013). A more model driven approach has leveraged the analytical formula of the gradient when using specific nonlinearities and topologies in order to develop parametrization of the recurrent weights. This has led to various unitary reparametrizations of the recurrent weight (Arjovsky et al., 2016; Wisdom et al., 2016; Helfrich et al., 2018; Henaff et al., 2016; Jing et al., 2017; Mhammedi et al., 2017; Hyland & Rätsch, 2017; Jose et al., 2018). A soft version of such parametrization lies in regularization of the DNNs. This includes dropout applied to either the output layer (Pham et al., 2014) or hidden state (Zaremba et al., 2014), noisin (Dieng et al., 2018), zoneout (Krueger et al., 2017) and recurrent batch normalization (Cooijmans et al., 2017). Lastly, identity initialization of ReLU RNNs has been studied in Le et al. (2015) and Talathi & Vartak (2016). Our results complements prior works in that simply using random initial hidden state instead of zero initial hidden state and without changing the RNN structure also relieves the exploding gradient problem by regularization the potentially largest term in the recurrent weight gradient.

