# OpenReview forum: "A Max-Affine Spline Perspective of Recurrent Neural Networks"
_ICLR.cc/2019/Conference_

### Official Review · AnonReviewer1 · 2018-10-31
**Very promising paper, in particular regarding applications, yet I found that heavy notation not so well explained made it hard to read**

**Rating:** 6
**Confidence:** 3

**Review:**

This paper builds upon recent work by Balestriero and Baraniuk (ICML 2018) that concern max-affine spline opertaor (MASO) interpretation of a substantial class of deep networks. In the new paper a special focus is put on Recurrent Neural Networks (RNNs), and it is highlighted based on theoretical considerations leveraging the MASO and numerical experiments that in the case of a piecewise affine and convex activation function, using noise in initial hidden state acts as regularization.
Overall I was impressed by the volume of contributions presented throughout the paper and also I very muched like the light shed on important classes of models that turn out to be not as black box as they could seem. My enthouasiasm was somehow tempered when discovering that the MASO modelling here was in fact a special case of Balestriero and Baraniuk (ICML 2018), but it seems that despite this the specific contribution is well motivated and justified, especially regarding application results. Yet, the other thing that has annoyed me and is causing me to only moderately champion the paper so far is that I found the notation heavy, not always well introduced nor explained, and while I believe that the authors have a clear understanding of things, it appears to me that the the opening sections 1 and 2 lack notation and/or conceptual clarity, making the paper hard to accept without additional care. To take a few examples:
a) In equation (3), the exponent (\ell) in A and B is not discussed. On a different level, the term "S" is used here but doesn't seem to be employed much in next instances of MASOs...why?
b) In equation (4), sure you can write a max as a sum with an approxiate indicator (modulo unicity I guess) but then what is called Q^{(\ell)} here becomes a function of A^{(\ell)}, B^{(\ell)}, z^{(\ell-1)}...?
c) In proposition 1, the notation A_sigma is not introduced. Of course, there is a notation table later but this would help (to preserve the flow and sometimes clarify things) to introduce notations upon first usage...
d) Still in prop 1, braket notation not so easy to grasp. What is A[z]z?
e) Still in prop 1, recall that sigma is assumed piecewise-linear and convex?
f) In th1, abusive to say that the layer "is" a mapping, isn't it?
g) In Theorem 2, what is f? A generic term for a deterministic function?
Also, below the Theorem, "affine" or "piecewise affine"?
h) I found section 4 somehow disconnected and flow-breaking. Put in appendix and use space to better explain the rest?
i) Section 5 is a strong and original bit, it seems. Should be put more to the fore in abstract/intro/conclusion?

---

> ### Author Response · Authors · 2018-11-10
> **We have made significant improvements to notations.**
>
> We thank the reviewer for their careful reading and constructive suggestions. We agree that the MASO framework sheds new light on the inner workings of RNNs. We have made significant simplifications and revisions to the mathematical notation, particularly in Sections 1.1, 1.2, and 2, that should address most of your concerns. Below we respond to your specific questions.
>
> a) We removed the exponents \ell in Section 1.2.  The reason for not using S in the remainder of the paper is that each operation in an RNN cell is a MASO S, e.g., we could have written an RNN cell operation as z_t = S_cell ( x , z_{t-1} ) = S_sigma ( S_W * x + S_z_{t-1} * z_{t-1} + b  ), but this would make the notation a bit more confusing. Therefore, we only use the notation S to introduce the definition of MASO, and omit it in the remainder of the paper.
>
> b) Implicitly, yes, Q is dependent on the affine parameters A and B of the MASO and the region in which the input x belongs. Here is a bit more detail on Q: given the parameters A and B and the input x, the MASO calculates the output through the internal maximization mechanism of the max-affine splines (see Eqs. 4 and 5 of the updated paper). This process infers (for each output dimension k) the region $r_k$ in which the input x belongs to, and adapts the rows of A and entries of B (of the affine mapping) accordingly. This process is highlighted in the paper through the tensor Q, in which the region inferred by the max-affine splines are stored as one-hot vectors. Stacking these region selection vectors for all output dimensions (all max-affine splines) row-wise, we obtain the partition section matrix Q. We have added a discussion about Q in Section 1.2 and revised Section 3 to make the explanation much cleaner.
>
> c) We have included the notation for A_\sigma in Proposition 1.
>
> d) We agree that the bracket notation is nonideal. The notation A[z]z is intended to indicate that the matrix A depends on the value of z (actually the partition region into which z falls). In an attempt to clarify the notation, in our revised paper, we use brackets strictly to denote matrix/vector value selection or concatenation. For example, [x]_k denotes the value of the k-th entry of the vector x, and [x_1, …, x_n] denotes the concatenation of the vectors x_1, …, x_n. Accordingly, we have omitted the input-dependency of the affine parameters. Instead, we make a note on page 3 to remind the reader that all affine parameters are input-dependent even though they are not explicitly written as such.
>
> e) We have added a footnote in the statement of Proposition 1 to reflect that \sigma is assumed to be piecewise affine and convex.
>
> f) Yes, we have unified our notation and now both a layer of an RNN and the overall RNN are referred to as a “piecewise affine spline operator” in their MASO formulation.
>
> g) “f” here denotes the RNN function, where the input is the concatenated input sequence and the output is the concatenated hidden states at the last layer. We have removed “f” in Theorem 2 to make it cleaner.
>
> h) We have made significant revisions and simplifications to the notation that hopefully improve the flow of the paper. Since Section 4 is an important section that contains the matched filterbank view of an RNN, we have kept this section in the main text.
>
> i) We have added a short overview of our contributions, including the noisy initial hidden state, in the second paragraph of the Introduction.
>
> Please let us know if the above address your concerns and if you have further inquiries.

---

### Official Review · AnonReviewer3 · 2018-11-02
**Interesting view point, but with limited applications**

**Rating:** 6
**Confidence:** 3

**Review:**

In this paper, the authors provide a novel approach towards understanding
RNNs using max-affine spline operators (MASO). Specifically, they rewrite RNNs
with piecewise affine and convex activations MASOs and provide some
explanation to the use of noisy initial hidden state.

The paper can be improved in presentation. More high level explanation should
be given on MASOs and why this new view of RNN is better.

To best of my knowledge, this is the first paper that related RNNs with MASOs
and provides insights on this re-formulation. However, the authors failed to
find more useful applications of this new formulation other than finding that
noisy initial hidden state helps in regularization. Also, the re-formulation
is restricted to piecewise affine and convex activation functions (Relu and
leaky-Relu).

In general, I think this is an original work providing interesting viewing
point, but could be further improved if the authors find more applications of
the MASO form.

---

> ### Author Response · Authors · 2018-11-10
> **Addressing concerns on limited applications of the MASO perspective of RNNs**
>
> We thank the reviewer for their constructive comments and suggested edits. We address each of them below.
>
> 1) Lack of application of the MASO formulation:
> In addition to improving the performance of RNNs using our suggestion of a noisy initial hidden state, our paper provides two additional insights/applications: (i) visualizing the progression through time of the RNN MASO input space partitioning and (ii) interpreting an RNN as a template matching machine (matched filterbank). These two applications are detailed in Sections 3 and 4; they provide new ways to visualize and interpret RNNs that complement related prior work on RNN visualization and interpretation.
>
> Future research directions and applications include the following, which have been added to the Conclusions of the paper (see Section 6). We can study whether enforcing an orthogonality constraint on the slope parameter A improves RNN performance, similar to what has been observed in [1] for deep feedforward networks. We can use the recently developed random matrix theory of deep learning [2] to analyze the affine slope parameter A (e.g., study how the distribution of its singular values changes during training) to analyze the implicit regularization that the optimizer performs when training RNNs.
>
> 2) Limitation of the analysis to convex activation functions:
> First, we acknowledge that focusing on piecewise affine and convex nonlinearities in RNNs might be limiting, since more elaborate models like LSTM and GRU use sigmoid and hyperbolic tangent activations. Nevertheless, having a solid understanding piecewise affine and convex nonlinearities in RNNs will guide subsequent theoretical development on other nonlinearities used in RNNs. Moreover, ReLU RNNs have recently gained considerable attention due to their simplicity, competitive performance, and ability to combat the exploding gradient problem provided they are parametrized and initialized properly. We have added a concise discussion in the third paragraph of the Introduction about ReLU RNNs to provide additional motivation for our work. In a future work direction, we expect that we can extend our convex/affine analysis to non-convex nonlinearities like the sigmoid and hyperbolic tangent by leveraging the development of the recent paper [3], which extend the MASO framework to more general nonlinearities. This development, however, is beyond the scope (and available space) of the current paper.
>
> Please let us know if the above address your concerns and if you have further inquiries.
>
> [1] Mad Max: Affine Spline Insights into Deep Learning (Balestriero and Baraniuk, 2018), https://arxiv.org/abs/1805.06576
> [2] Implicit Self-Regularization in Deep Neural Networks: Evidence from Random Matrix Theory and Implications for Learning (Martin and Mahoney, 2018), https://arxiv.org/abs/1810.01075
> [3] From Hard to Soft: Understanding Deep Network Nonlinearities via Vector Quantization and Statistical Inference (Balestriero and Baraniuk, 2018), https://arxiv.org/abs/1810.09274

---

### Official Review · AnonReviewer2 · 2018-11-02
**Good quality paper, experimentation can be improved**

**Rating:** 6
**Confidence:** 3

**Review:**

The paper rewrites equations of Elman RNN in terms of so-called max-affine spline operators. Paper claims that this reformulation allows better analysis and, in particular, gives an insight to use initial state noise to regularize hidden states and fight exploding gradients problem.

The paper seems to be theoretically sound. The experiment with sequential MNIST looks very promising, thought it would be great to check this method on other datasets (perhaps, toy data) to check that this is not a fluke. The bird audio dataset is not well benchmarked in the literature. The paper could make much stronger claim with more extensive experimentation.

Some typos:
- p3: an simple -> a simple
- Figure 2 caption is not finished
- p5 last paragraph: extra full stop
- Fig 3: correct and negative probably switched around
- p7: in regularize -> in regularization

---

> ### Author Response · Authors · 2018-11-10
> **Addressing concerns on experiments and datasets**
>
> We thank the reviewer for their constructive comments. First, all the typos have been corrected in the updated manuscript. Second, we have made significant simplifications to the mathematical notation in Sections 1 and 2 that improve the clarity of presentation. We address the remaining concerns below.
>
> 1) Regarding the seemingly insufficient experimental evaluation:
> We actually evaluated the use of the noise in the initial hidden state on not one but four datasets of four different modalities: simulated toy data (artificial), MNIST (imagery), SST-2 (text), and bird detection (audio). Our goal (which was achieved) was to demonstrate that, for simple RNNs, injecting noise into initial hidden state improves performance for all four modalities. We present additional successful experimental results in Appendix F of the Supplementary Material; we could not include these in the main text due to space limitations. Our experiments on these four datasets/modalities provide strong evidence on the utility of the noisy initial hidden state. Additional results/visualizations of the input space partitioning and matched filtering are included in Appendices D and E, respectively.
>
> For the exploratory experiments (last part in Section 5.2), we have added experimental results on MNIST and permuted MNIST datasets using a one-layer GRU that similarly demonstrates the potential gain in classification accuracy when using noisy initial hidden state in more complex models where nonlinearities are no longer piecewise affine and convex.
>
> 2) Regarding the Bird Detection Dataset being not well benchmarked:
> This dataset is, in fact, well benchmarked; perhaps we failed to make it clear in the main text. Please see this website for the task description (http://machine-listening.eecs.qmul.ac.uk/bird-audio-detection-challenge) and this website for a list of benchmarks (http://c4dm.eecs.qmul.ac.uk/events/badchallenge_results). We have included the link to the benchmarks in the main text; see the new footnote on page 8.
>
> Please let us know if the above address your concerns or if you have further inquiries.

---

### Meta-Review · Area_Chair1 · 2018-12-18
**Reasonable strong theory paper**

**Confidence:** 4
**Recommendation:** Accept (Poster)

**Metareview:**

While the reformulation of RNNs is not practical as it is missing sigmoids and tanhs that are common in LSTMs it does provide an interesting analysis of traditional RNNs and a technique that's novel for many in the ICLR community.